# Prevalence of Congenital Infections in Newborns and Universal Neonatal Hearing Screening in Santa Catarina, Brazil

Eduarda Besen [1], Karina Mary Paiva [1], Luciana Berwanger Cigana [2], Marcos José Machado [3], Alessandra Giannella Samelli [4] and Patrícia Haas [1,*]

1 Department of Speech Therapy, Federal University of Santa Catarina, Florianópolis 88040-970, Brazil
2 Voice, Speech and Language Hearing Clinic, Instituto Otovida, Capoeiras 88085-002, Brazil
3 Clinical Analysis Department, Federal University of Santa Catarina, Florianópolis 88040-970, Brazil
4 Faculdade de Medicina, Universidade de São Paulo—FMUSP, São Paulo 01246-903, Brazil
* Correspondence: patrícia.haas@ufsc.br; Tel.: +55-48-999614949

**Abstract:** Objective: to verify the frequency of congenital infections in newborns and their possible associations with the universal-neonatal-hearing-screening (UNHS) results, and evaluate a reference UNHS service in the Unified Health System (*Sistema Único de Saúde*—SUS), according to quality indicators. Methods: Historical cohort study with data analysis of newborns attending prestigious hearing-health SUS services from January 2017 to December 2021, in Santa Catarina, Brazil. The quality of screening coverage was assessed based on the quality indicators proposed by the Brazilian neonatal-hearing-screening-care guidelines (*Diretrizes de Atenção da Triagem Auditiva Neonatal*—DATAN). Logistic-regression analysis, crude OR calculations, Cochran–Mantel–Haenszel OR calculation, and chi-square test were performed to estimate the association between risk indicators for hearing loss and UNHS failure. Results: In the last five years, the prestigious services performed UNHS on 34,801 newborns and met the DATAN quality indicators. Congenital syphilis was the most frequent (1.59%) congenital infection in newborns, followed by HIV (0.87%), whereas the least frequent was rubella (0.029%). Conclusion: Prestigious UNHS services reached ≥95% hearing screening coverage. Considering all congenital infections, the prevalence was 2.87%, with congenital syphilis the most frequent. Newborns with congenital syphilis or HIV are more likely to fail UNHS.

**Keywords:** hearing; communicable diseases; risk indicator; public policy; neonatal screening

## 1. Introduction

One of the public policies implemented by the Ministry of Health of Brazil [1] is called the "Stork Network", which provides healthcare to pregnant women and newborns. It focuses on quality prenatal care and comprehensive health care for women and children from birth to 24 months, ensuring access, support, and problem-solving for them in prenatal care, childbirth, and puerperium [1–3]. Congenital infections can occur as microorganisms pass through the placenta or breast milk, or by contact with blood or vaginal discharges in the prenatal, perinatal, or postnatal periods [4,5], and are important causes of fetal and neonatal mortality, as well as developmental sequelae [6–8].

In the epidemiological scenario, congenital and perinatal infections (of which toxoplasmosis, congenital rubella, cytomegalovirus, congenital syphilis, and human immunodeficiency virus [HIV] are the most commonly observed in previous studies) are important risk indicators for hearing loss (RIHL) [9,10]. In Brazil, the 2017 incidence rate of congenital syphilis was 8.6/1000 live births [11], and the 2020 incidence rate was 7.7/1000 live births [12]. The vertical-HIV-transmission rate was 2.8/1000 live births in 2017 [13], and 2.7/1000 live births in 2020 [14]. Other incidence rates were as follows: congenital toxoplasmosis: 1:10/1000 [15]; cytomegalovirus: from 0.2 to 2.2% [16,17]; and congenital rubella [18], in outbreaks of congenital rubella syndrome: 4.3/1000 live births. This scenario

calls for strategies to prevent congenital infections and provide essential prenatal, perinatal, and postnatal mother and child health-care. One of the main aggravations of congenital infections is hearing loss in newborns and infants [19–22].

Furthermore, several studies and institutions, including Joint Committee on Infant Hearing (JCIH) guidelines, report that other RIHL in newborns include neonatal-intensive-care unit (NICU) stay of more than 5 days, duration of assisted ventilation, low Apgar scores, ototoxic drug exposure, craniofacial anomalies, and so forth [19–25].

Universal neonatal hearing screening (UNHS), which is essential to the early detection of hearing loss, also provides comprehensive, pediatric hearing-health-care with hearing and language monitoring and follow-up, diagnosis, and (re)habilitation. The Care Network for People with Disabilities recommends using it to organize healthcare, focusing on the needs of people with hearing loss at different levels of complexity of the Unified Health System (*Sistema Único de Saúde*—SUS) [10,21].

UNHS programs may adopt the following scientific institutions' protocols: JCIH [19,20], the multiprofessional committee on hearing health (*Comitê Multiprofissional de Saúde Auditiva*—COMUSA) [21], and the neonatal-hearing-screening-care guidelines (*Diretrizes de Atenção da Triagem Auditiva Neonatal*—DATAN) [22] to determine quality indicators of hearing-loss identification, confirmation, diagnosis, and early rehabilitation, and thus control the effectiveness of the implemented program [23,24].

DATAN [22] recommends the following quality indicators to verify and monitor the effectiveness of UNHS programs in Brazil: (1) screening-coverage index (≥95%); (2) age in months at screening (up to the first month of life or the third month of life—corrected age—for premature infants in cases of hospitalization); (3) rate of referrals for diagnosis (2% to 4%); 4) rate of attendance at diagnosis (≥90%); (5) age at diagnosis (up to the third month of life); 6) speech therapy started in 95% of infants; (7) hearing-aid fitting within one month after diagnosis in 95% of diagnosed infants. Slightly more than a decade after UNHS was implemented, the program is not yet equally effective in all Brazilian regions, possibly due to distinct sociodemographic and cultural characteristics, difficulties in hiring professionals and maintaining adequate equipment and accessories, and UNHS registration limitations [25,26].

Given this context, this study aimed to estimate the prevalence of communicable diseases in newborns and their possible associations with UNHS results, and to evaluate a prestigious UNHS service in SUS, according to international quality indicators.

## 2. Materials and Methods

### 2.1. Design and Study Site

The study used data on UNHS regarding babies born in the Carmela Dutra Maternity Hospital (MCD) (Florianópolis, Santa Catarina [SC], Brazil) and the São José Regional Hospital (HRSJ) (São José—SC). UNHS is preferably performed in the first 24 to 48 h of life, in the maternity hospitals, or within 30 days from birth at the Otovida Institute (a prestigious hearing-health service in SC). Newborns are to be registered in the service database and evaluated with transient-evoked otoacoustic emission (TEOAE) in both ears and/or automated auditory brainstem response (AABR) according to RIHL [20,21]. In the presence of RIHL, the newborns who "passed" the AABR were referred for hearing monitoring in primary care, and those who "failed" were referred for retesting in the state outpatient hearing-health service. According to the protocol followed in Brazil (DATAN [22]), newborns who have RIHL and fail the screening are retested once more outside the hospital with the AABR and, if the failure persists, they undergo a complete audiological evaluation for audiological diagnosis. All stages of screening and audiological evaluation are carried out by audiologists and, if it is necessary to confirm the diagnosis of hearing loss, a multidisciplinary team is involved, including the otorhinolaryngologist.

### 2.2. Screening and Data-Collection Procedure

Data were analyzed based on the Otovida Institute database, a prestigious hearing-health service in SC responsible for conducting UNHS in public maternity hospitals (MCD and HRSJ). Information about the following aspects was collected: prenatal care, childbirth, puerperium, the mothers' and newborns' sociodemographic characteristics, TEOAE and/or AABR test results (satisfactory "PASS" or unsatisfactory "FAIL"), and RIHL (family history of permanent deafness; consanguinity; NICU stay for more than five days; use of extracorporeal ventilation, assisted ventilation, and ototoxic drugs (such as aminoglycoside antibiotics and/or loop diuretics); hyperbilirubinemia, severe perinatal anoxia, 1-min Apgar score of 0 to 4, or 5-min score of 0 to 6, birth weight less than 1500 g, communicable diseases (infectious diseases), craniofacial anomalies involving the ear and temporal bone, genetic syndromes that usually cause disabilities, and neurodegenerative disorders).

UNHS data were analyzed to evaluate the quality of the service, as proposed by DATAN [22], including UNHS performed within 30 days of life. The analysis also addressed the percentage of those screened of the number of live births, obtained from the websites of the Ministry of Health, TabNet (DATASUS—visits performed) and the State Department of Health regarding the number of visits in MCD and HRS (screenings performed).

### 2.3. Outcome Variable

UNHS, categorized as "pass" or "fail", was assessed as the outcome variable. Newborns who failed the TEOAE and/or AABR in only one or both ears were considered "fail".

### 2.4. Main Exposure Variable and Covariates

The main research variables were toxoplasmosis, congenital rubella, cytomegalovirus, congenital syphilis, herpes, and HIV (no or yes). The covariates consisted of year of birth (2017; 2018; 2019; 2020; 2021), maternal age ($\leq$19 years; 20 to 29 years; $\geq$30 years), RIHL (NICU stay for more than five days, antibiotic use, low Apgar scores, use of mechanical ventilation and/or blood transfusion, prematurity, craniofacial anomalies and/or neurological disorders, family history of hearing loss) (no or yes). The prestigious hearing-health service controlled the tests to meet UNHS quality parameters [27].

### 2.5. Data Analysis

The data were organized in Microsoft Excel® spreadsheets and then exported to and analyzed in StataMP® software, version 14.0 (StataCorp, College Station, TX, USA). Descriptive analyses were presented with absolute and relative frequencies and their 95% confidence intervals (95% CI). The association between UNHS failure (outcome) and communicable diseases (main exposure) and research covariates was analyzed. The odds ratio (OR) was used as a measure of association for both the crude (bivariate) and adjusted analysis, estimated with logistic-regression analysis.

Subsequently, the data were organized in Microsoft Excel spreadsheets and then exported to and analyzed in MedCalc® Statistical Software, version 20.027. The categorical variables of the sample were described in absolute and relative frequencies, with their 95% confidence intervals (95% CI). The association between the outcome (UNHS) and the main exposure (toxoplasmosis, congenital rubella, cytomegalovirus, congenital syphilis, herpes, and HIV) and research covariates was analyzed with the chi-square test, which, when possible, was also applied to evaluate trends (year of birth and categorized maternal-age). The OR was used as a measure of association in the crude (bivariate) and adjusted analyses, estimated with logistic-regression analysis and 2 × 2 table calculations (crude OR) or with the Cochran–Mantel–Haenszel test. The program-effectiveness evaluation results were described according to DATAN quality indicators [22].

### 2.6. Ethical Aspects

This study was approved by the Research Ethics Committee. CAAE: 39562720.8.0000.0121.

## 3. Results

Regarding data analysis of the prestigious services of SC, the coverage in both MCD and HRSJ was greater than the recommended 95% (Table 1).

**Table 1.** Analysis of the number of visits (screenings performed) and percentage of those screened of the number of live births, obtained from the websites of the Ministry of Health, TabNet (DATASUS—visits performed), and the State Department of Health in the two maternity hospitals. Florianópolis, SC (2017 to 2021).

| Year | Data from the SC Department of Health | Data from DATASUS | Data from the SC Department of Health | Data from DATASUS | Data from the SC Department of Health | Data from DATASUS |
|---|---|---|---|---|---|---|
| Health Facility | MCD * Florianópolis, SC | | HRSJ ** São José, SC | | Percentage of Otovida Coverage—Overall | Percentage of Otovida Coverage—Overall |
| 2017 | 98.01% | 98.43% | 89.48% | 87.46% | 93.82% | 92.97% |
| 2018 | 109.38% | 110.24% | 89.16% | 90.28% | 99.36% | 100.37% |
| 2019 | 101.96% | 102.48% | 106.44% | 105.92% | 104.19% | 104.20% |
| 2020 | 97.04% | 97.46% | 100.37% | 105.55% | 98.72% | 101.46% |
| 2021 | 110.35% | 114.01% | 100.23% | 101.90% | 105.02% | 107.58% |
| Total | 103.35% | 104.42% | 96.79% | 97.72% | 100.56% | 101.05% |
| Annual average | 103.35% | 104.52% | 97.13% | 98.22% | 100.22% | 101.32% |
| Overall annual average | | | | | 100.77% | |

\* MCD: Carmela Dutra Maternity Hospital; HRSJ \*\*: São José Regional Hospital Dr Homero de Miranda Gomes.

UNHS data regarding 34,801 patients were analyzed, and only 1.106% (95% CI 1.001–1.226%) of the newborns were referred for retesting at the prestigious service, due to hearing-screening failure.

Congenital syphilis was the most frequent (average 1.59%) congenital infection in newborns who underwent UNHS between 2017 and 2021, followed by HIV (average of 0.87%). The least frequent was rubella (average of 0.029%) (Figure 1).

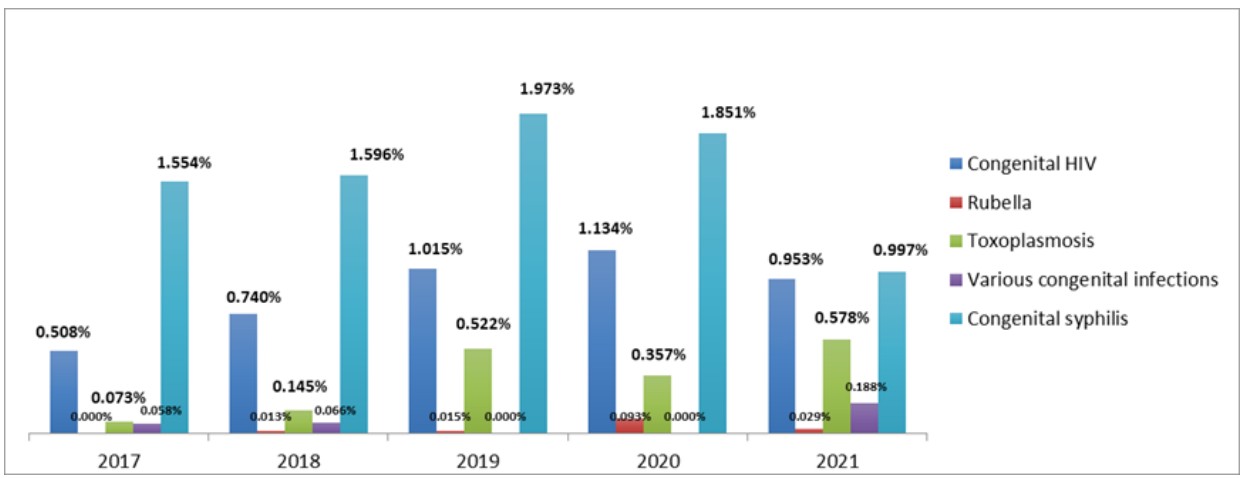

**Figure 1.** Relative frequencies of congenital infections in newborns who underwent UNHS, by year of birth. Florianópolis, SC, 2017 to 2021 (n = 34,801).

The RIHL varied between newborns (Figure 2) and year of birth, indicating differences and even trends of increase or decrease from year to year. Prematurity, NICU stay, and antibiotic use were the most frequent RIHL between 2017 and 2021. Concerning craniofacial anomalies in conjunction with neurological disorders, the combined frequency of all years was 0.199%, but these frequencies showed a significant increase over the years (Table 2).

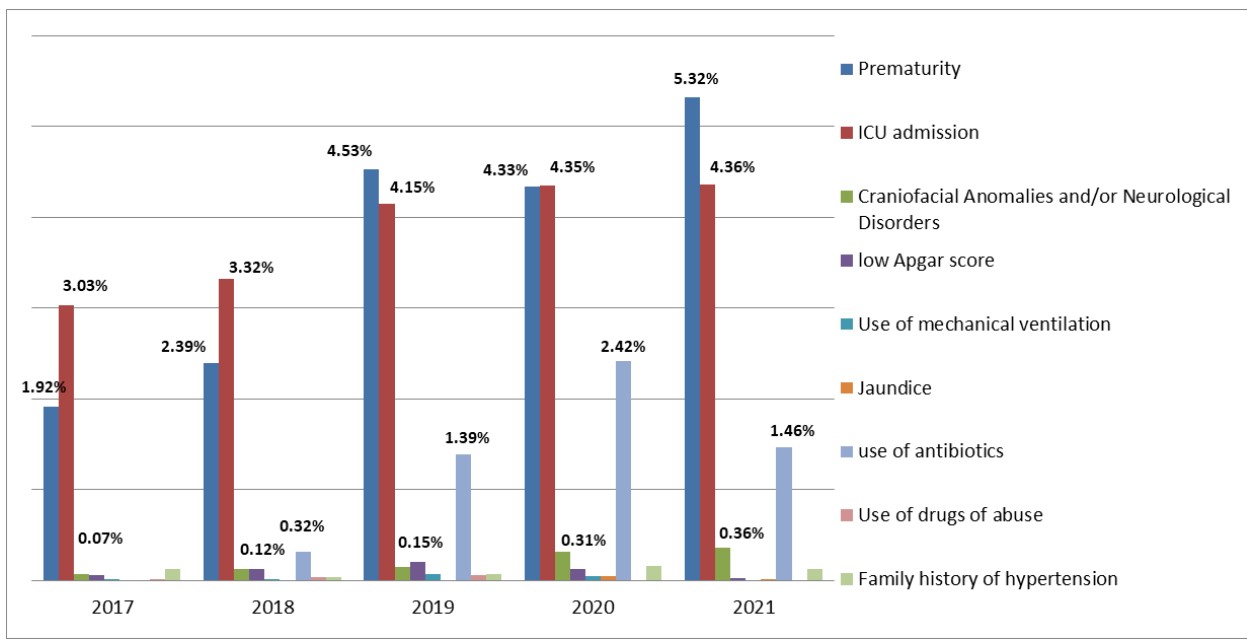

**Figure 2.** Relative frequencies of RIHL in newborns who underwent UNHS, by year of birth. Florianópolis, SC, 2017 to 2021 (n = 34,801).

**Table 2.** Relative frequency of craniofacial anomalies and neurological disorders in newborns in UNHS, by year of birth. Florianópolis, SC, 2017 to 2021 (n = 34,801).

| Variable | Year of Birth | n | % | 95% CI | *p*-Value * | *p*-Value ** | *p*-Value *** |
|---|---|---|---|---|---|---|---|
| Craniofacial Anomalies | 2017 | 5 | 0.0726 | 0.0236 to 0.169 | 0.1113 | 0.0148 | 0.1877 |
| | 2018 | 8 | 0.106 | 0.0456 to 0.208 | | | |
| | 2019 | 10 | 0.145 | 0.0696 to 0.267 | | | |
| | 2020 | 15 | 0.233 | 0.130 to 0.384 | | | |
| | 2021 | 13 | 0.188 | 0.100 to 0.321 | | | |
| | Total | 51 | 0.147 | 0.109 to 0.193 | | | |
| Neurological Disorders | 2017 | 0 | 0 | 0.000 to 0.0535 | <0.0001 | <0.0001 | 0.0057 |
| | 2018 | 1 | 0.0132 | 0.000334 to 0.0736 | | | |
| | 2019 | 0 | 0 | 0.000 to 0.0535 | | | |
| | 2020 | 5 | 0.0776 | 0.0252 to 0.181 | | | |
| | 2021 | 12 | 0.173 | 0.0896 to 0.303 | | | |
| | Total | 18 | 0.0519 | 0.0307 to 0.0819 | | | |

95% CI: 95% confidence interval. *p*-value * chi-square test; *p*-value ** chi-square trend test; *p*-value *** chi-square test of proportions.

There were mostly no significant differences between the frequencies of newborns with RIHL who failed and those who passed UNHS. The 0.09 OR shows that newborns born in 2021 were approximately 91% less likely to fail UNHS than those born in 2017. However, HIV-positive newborns were 388% to 1143% more likely to fail the UNHS (Table 3).

**Table 3.** Odds ratio adjusted using the Cochran–Mantel–Haenszel method for the association between UNHS failure, newborns with RIHL, or mother-related variables. Florianópolis, SC, 2017 to 2021 (n = 34,801).

| Variable | Adjusted OR | 95% CI | *p* Value |
|:---:|:---:|:---:|:---:|
| Congenital HIV | | | |
| No | 1.0000 | 3.8848 to 11.4375 | <0.0001 |
| Yes | 6.6658 | | |
| Congenital syphilis | | | |
| No | 1.0000 | 1.4465 to | 0.0006 |
| Yes | 2.3759 | 3.9024 | |
| Craniofacial Anomalies and/or Neurological Disorders | | | |
| No | 1.0000 | 21.9068 to 89.2331 | <0.0001 |
| Yes | 44.2133 | | |
| Admission to the ICU | | | |
| No | 1.0000 | 2.1486 to | <0.0001 |
| Yes | 3.1244 | 4.5434 | |
| Prematurity | | | |
| No | 1.0000 | 3.6051 to | <0.0001 |
| Yes | 5.0031 | 6.9432 | |
| Antibiotic use | | | |
| No | 1.0000 | 1.0092 to | 0.0474 |
| Yes | 2.2419 | 4.9801 | |
| Year of Birth | | | |
| 2017 | 1.0000 | | |
| 2018 | 1.1525 | 0.8585 to 1.5472 | 0.3449 |
| 2019 | 1.7179 | 1.3017 to 2.2671 | 0.0001 |
| 2020 | 0.3440 | 0.2251 to 0.5257 | <0.0001 |
| 2021 | 0.0900 | 0.0449 to 0.1803 | <0.0001 |
| Maternal age | | | |
| X | 1.0000 | 0.9672 to | 0.0279 |
| x + 1 year | 0.9825 | 0.9981 | |

95% CI: 95% confidence interval; OR: odds ratio.

## 4. Discussion

From 2017 to 2021, the UNHS was performed on 34,801 newborns in two maternity hospitals, representing 100.75% of the 34,720 live births, according to data from the SC State Department of Health. The number of screened newborns exceeds the number of live births in the two maternity hospitals in these years, which may be due to incomplete information, delay in updating the number of newborns on the official Brazilian birth notification websites, and newborns who were admitted to the NICU [28] of the surveyed maternity hospitals, although born in other ones.

It can be verified that UNHS coverage is efficient, according to DATAN [22]. Results above 95% coverage were also achieved by other UNHS programs in Brazil [26,29,30], although some programs have not yet reached this coverage index [31,32], especially those that did not perform screening before hospital discharge [33–35]. It is important to emphasize that in March 2020, when a lockdown due to COVID-19 was decreed, the hearing-loss detection phase only occurred in an outpatient setting at the prestigious service in UNHS for SUS. During this period, the maternity and primary-care teams emphasized the importance of performing the UNHS, avoiding the evasion of screening. In April, the audiologists returned to perform the screening before hospital discharge in the two maternity hospitals, as recommended by COMUSA [36]. Thus, we believe that the rate of loss of UNHS during this period was minimal.

The literature shows that the frequency of hearing loss is higher in newborns with RIHL [22,23,36,37]. Those evaluated in the present study had different RIHL, with differences in frequencies from year to year, and even trends of increase or decrease. It was observed that the most common RIHL in 2021 among newborns were prematurity (5.31%),

NICU stay longer than five days (4.36%), and ototoxic drug use (1.45%)—these frequencies are similar to a previous study [38]. If we consider the 5-year average frequency, we have 3.81% for NICU stay and 3.65% for prematurity. The least observed RIHL in the present study were craniofacial anomalies [39] and/or neurological disorders, with a 5-year combined frequency of 0.199%. However, the adjusted OR showed a significant association between having this RIHL and being more likely to fail the UNHS.

Another important and common RIHL is congenital infections [40]. Considering all infections together (toxoplasmosis, congenital rubella, cytomegalovirus, congenital syphilis, herpes, HIV, and other congenital infections) the present study found an average frequency between 2017 and 2021 of 2.87%, the third major RIHL. Previous studies report different RIHL, which may be explained by different conditions such as specific types of NICU, countries, populations, etc. [41] In the adjusted OR analysis, we observed a significant association between having congenital syphilis or congenital HIV and being more likely to fail the UNHS.

Our hypothesis regarding the fewer cases of congenital syphilis in 202 is that it is probably related to the public policies that increase primary prevention measures. These include the effectiveness of the multidisciplinary team and the interpretation of VDRL tests, primarily comparing exposed newborns with their mothers, after delivery.

Despite the prevalence of risk indicators observed in the present study, only 1.1% of the newborns failed the hearing screening. Most of the time, no significant differences were observed between the frequencies of newborns with RIHL who failed and those who passed the UNHS, but, regarding infectious diseases, there was a higher proportion of failure in neonates when compared to neonates without infectious diseases, agreeing with findings of a previous study [38]. However, since the collected data refer to a single moment in the hearing-health program, that is, the UNHS, it is not possible to know whether the diagnosis of hearing loss was confirmed. Due to the presence of RIHL, hearing loss can develop later in many cases and, therefore, the longitudinal follow-up of these children is fundamental. In addition, those newborns who have failed the UNHS should be retested as recommended [19–22]. Even those who have passed the UNHS but have RIHL must be monitored in terms of hearing and language development [10,21].

## 5. Conclusions

The prestigious UNHS service where this study was conducted achieved a screening coverage ≥95%.

Considering all congenital infections (toxoplasmosis, congenital rubella, cytomegalovirus, congenital syphilis, herpes, HIV, other congenital infections), the prevalence of RIHL was 2.87%, with congenital syphilis being the most frequent.

Newborns with congenital syphilis or HIV are more likely to fail the UNHS.

The knowledge produced in this study is expected to raise awareness of health professionals' performance in the three levels of health care, thus promoting comprehensive pediatric care, expanding access to health care, and implementing and strengthening UNHS to ensure continuity of care.

**Author Contributions:** Conceptualization, K.M.P. and P.H.; formal analysis, E.B., K.M.P., L.B.C., M.J.M., A.G.S. and P.H.; data collection, E.B. and L.B.C.; writing—review and editing, E.B., K.M.P., L.B.C., M.J.M., A.G.S. and P.H. All authors have read and agreed to the published version of the manuscript.

**Funding:** This research received no external funding.

**Institutional Review Board Statement:** The study was conducted in accordance with the Declaration of Helsinki and approved by the Research Ethics Committee. CAAE: 39562720.8.0000.0121.

**Informed Consent Statement:** Not applicable.

**Data Availability Statement:** Not applicable.

**Conflicts of Interest:** The authors declare no conflict of interest.

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
