# Peer review of "Prevalence of Congenital Infections in Newborns and Universal Neonatal Hearing Screening in Santa Catarina, Brazil"

_audiolres, doi:10.3390/audiolres13010011_

Round 1

Reviewer 1 Report

Please see file attached

Author Response

COVER LETTER

We would like to thank you for the opportunity to resubmit the revised version of our manuscript entitled “PREVALENCE OF CONGENITAL INFECTIONS IN NEWBORNS AND UNIVERSAL NEONATAL HEARING SCREENING IN SANTA CATARINA, BRAZIL” for consideration for publication in the Audiology Research. We are thankful to the referees and the Editor for their thoughtful suggestions that helped to strengthen our manuscript. This new version of the manuscript included changes in the manuscript following reviewers’ recommendations. We have addressed all the reviewers’ concerns and provided a detailed response below. To facilitate the process, we have transcribed all questions raised and answered them point by point. In addition, we are attaching a marked version of the manuscript.

Please do not hesitate to contact us if you have any further questions. We look forward to hearing back from you soon.

REVIEWER 1

Open Review

English language and style

( ) English very difficult to understand/incomprehensible
(x) Extensive editing of English language and style required
( ) Moderate English changes required
( ) English language and style are fine/minor spell check required
( ) I don't feel qualified to judge about the English language and style

Yes

Can be improved

Must be improved

Not applicable

Does the introduction provide sufficient background and include all relevant references?

( )

(x)

( )

( )

Are all the cited references relevant to the research?

( )

(x)

( )

( )

Is the research design appropriate?

( )

( )

(x)

( )

Are the methods adequately described?

( )

( )

(x)

( )

Are the results clearly presented?

( )

( )

(x)

( )

Are the conclusions supported by the results?

( )

(x)

( )

( )

Communicable Diseases In Newborns And Universal Neonatal Hearing  Screening Coverage Analysis

The topic is interesting

I have  few remarks.

Title and text

I would change the title, eliminating coverage analysis. This topic has been analysed many many times in the literature. Prevalence of communicable diseases in newborns  is much more important.

Response: Thank you. We appreciate your positive feedback. Thank you for the opportunity to submit the revised version of our manuscript. We have addressed all the comments, including the changing in the title. We also included the suggestion of the another reviewer.

Introduction

Lines 40-42 In the epidemiological scenario, congenital and perinatal infections are prevalent such as toxoplasmosis, congenital rubella, cytomegalovirus, congenital syphilis, herpes, and  HIV [9,10]

Do you mean that toxo, etc are prevalent on others infections or that congenital infections are widespread ?This sentence is not so clear, to me.

Response: Thank you. We corrected the sentence.

Lines 62-65 UNHS programs may adopt the following protocols from the scientific institutions Joint Committee on Infant Hearing (JCIH)[19,20], Multiprofessional Committee on Auditory Health (COMUSA)[21], and Neonatal Hearing Screening Care Guidelines (DATAN)[22]  to determine quality indicators for the phases of identification, confirmation, diagnosis,  and early rehabilitation of hearing loss to control the effectiveness of the implemented  program[23,24]. The UNHS service in the State of Santa Catarina (SC) is based on 64 DATAN[20,22].

What  does it mean?That each region of Brazil has the possibility to choose a protocol?Why the state of Santa Catarina is cited in this context? May be this sentence is pleonastic if the all state follows DATAN...

Response: Thank you. You are right. We excluded the sentence.

Lines 77-80 Given this context, this study aimed to estimate the prevalence of communicable  diseases in newborns and their possible associations with the results of UNHS, in addition to evaluating a reference service in UNHS for SUS according to international  quality indicators.

The same remark concerning the  title. You can just report the  results and quality indicators without a specific discussion about the screening in general.

 Of course this is my opinion but I think that  quality analys should be object of another  paper, better in a local journal.

Response: Thank you. We will consider your opinion in discussion section.

Methods

Lines 83-88

Secondary data from a reference service in UNHS were used, with screening in  newborns who were born in the Carmela Dutra Maternity Hospital – (MCD)  (municipality of Florianópolis - SC) and in the maternity of the Regional São José  Hospital – (HRSJ) (municipality of São José - SC) and who performed the UNHS  preferably in the first hours of life, 24h to 48h, in the maternity hospitals or up to 30 days  of birth at the Otovida Institute (Clinic of hearing, voice, speech, and language, a service  accredited to SUS, a reference for Hearing Health in the State of SC)

Sentence too long and  not  clear

Response: Thank you. We rewrite the sentence.

Results

The results are mainly focused on hearing risk factors  but it is not  clear what happened after the screening. Was hearing loss always confirmed? In all cases? The entity?

Response: Thank you for your observation. As our study include data at the moment of the hearing screening, the diagnosis of hearing loss had not yet been closed and, in many cases, due to the risk indicator, hearing loss can develop later and, therefore, the longitudinal follow-up of these children it is of fundamental importance  (for newborns who failed the screenings or even for those who passed but who have risk indicators). We believe that what was highlighted by the reviewer is important and, therefore, we have added this item to the discussion.

I think that this paper has interesting data concerning congenital infections. I would concentrate the discussion  on this. How is the situation in other developing countries? Quality indicators of  NHS is a old topic and this article, in this way, does not add anything to the problem.

 Of course , this is my opinion...

Response: Thank you for your observation. We made some changes in Discussion section trying to emphasize this topic.

Reviewer 2 Report

Title: it seems to be more correct to use definition “congenital infections” instead of “communicable diseases” and indicate the site of the study (ex. “Congenital infections and universal neonatal hearing screening coverage analysis in the state of Santa Catarina, Brazil”

Abstract: very detailed methods section but results are very scarce. The exact value of UNHS coverage indicator should be provided here as the wide analysis is provided in the main text and the coverage outcome is mentioned in the title. Another issue the risk indicators for hearing loss: a wide analysis was performed and presented in the main text but its results aren’t mentioned in the abstract.

Introduction: All quality indicators of the UNHS guideline are listed but the title and results are only about one outcome – the screening coverage. Redundant information makes it difficult to understand the main idea.

Line 68 – 2) age in months at screening – a mistake? age in days?

Methods: very well described.

Results and discussion: During the first reading it was quite challenging to associate English full-named definitions with Portuguese abbreviations (SUS, DATAN) introduced earlier.

The chart 1 is actually organized as a table so it must be labeled as a table with changing numbers of the next tables.

Please present figure 1 and 2 in English and colored. The figure 2 would be perceived better being organized as figure 1, with yearly data grouped closely in each RIHL category.

Concerning all provided data – an extensive yearly analysis of main variables and covariates was performed and presented to reveal some trends. But the interpretation is quite brief in the results as well as in the discussion. It doesn't clear why  a definite table is dedicated to craniofacial anomalies and/or neurological disorders despite their rather low prevalence.  The congenital infections are emphasized in the title, introduction and the aim of the study, but there is no word on them in the discussion. More concordance is expected.

Author Response

COVER LETTER

We would like to thank you for the opportunity to resubmit the revised version of our manuscript entitled “PREVALENCE OF CONGENITAL INFECTIONS IN NEWBORNS AND UNIVERSAL NEONATAL HEARING SCREENING IN SANTA CATARINA, BRAZIL” for consideration for publication in the Audiology Research. We are thankful to the referees and the Editor for their thoughtful suggestions that helped to strengthen our manuscript. This new version of the manuscript included changes in the manuscript following reviewers’ recommendations. We have addressed all the reviewers’ concerns and provided a detailed response below. To facilitate the process, we have transcribed all questions raised and answered them point by point. In addition, we are attaching a marked version of the manuscript.

Please do not hesitate to contact us if you have any further questions. We look forward to hearing back from you soon.

Reviewer 2

Open Review

English language and style

( ) English very difficult to understand/incomprehensible
( ) Extensive editing of English language and style required
( ) Moderate English changes required
( ) English language and style are fine/minor spell check required
(x) I don't feel qualified to judge about the English language and style

Yes

Can be improved

Must be improved

Not applicable

Does the introduction provide sufficient background and include all relevant references?

(x)

( )

( )

( )

Are all the cited references relevant to the research?

( )

(x)

( )

( )

Is the research design appropriate?

(x)

( )

( )

( )

Are the methods adequately described?

(x)

( )

( )

( )

Are the results clearly presented?

( )

( )

(x)

( )

Are the conclusions supported by the results?

( )

( )

(x)

( )

Comments and Suggestions for Authors

Title: it seems to be more correct to use definition “congenital infections” instead of “communicable diseases” and indicate the site of the study (ex. “Congenital infections and universal neonatal hearing screening coverage analysis in the state of Santa Catarina, Brazil”

Response: Thank you. We appreciate your positive feedback. Thank you for the opportunity to submit the revised version of our manuscript. We have addressed all the comments, including the changing in the title. We also included the suggestion of the another reviewer.

Abstract: very detailed methods section but results are very scarce. The exact value of UNHS coverage indicator should be provided here as the wide analysis is provided in the main text and the coverage outcome is mentioned in the title. Another issue the risk indicators for hearing loss: a wide analysis was performed and presented in the main text but its results aren’t mentioned in the abstract.
Response: Thank you. We reformulated the abstract; we give less emphasis on coverage and more emphasis on the prevalence of congenital infections and failures in the UNHS.

Introduction: All quality indicators of the UNHS guideline are listed but the title and results are only about one outcome – the screening coverage. Redundant information makes it difficult to understand the main idea.

Response: Thank you: We rewrote many parts of the text trying to make it clearer.

Line 68 – 2) age in months at screening – a mistake? age in days?

Response: Thank you: We rewrote this sentence.

Methods: very well described.

Response: Thank you.

Results and discussion: During the first reading it was quite challenging to associate English full-named definitions with Portuguese abbreviations (SUS, DATAN) introduced earlier.
Response: Thank you. We included the acronyms in Portuguese.

The chart 1 is actually organized as a table so it must be labeled as a table with changing numbers of the next tables.

Response: Thank you. We made this change.

Please present figure 1 and 2 in English and colored. The figure 2 would be perceived better being organized as figure 1, with yearly data grouped closely in each RIHL category.

Response: Thank you. We made this change.

Concerning all provided data – an extensive yearly analysis of main variables and covariates was performed and presented to reveal some trends. But the interpretation is quite brief in the results as well as in the discussion. It doesn't clear why definite table is dedicated to craniofacial anomalies and/or neurological disorders despite their rather low prevalence.  The congenital infections are emphasized in the title, introduction and the aim of the study, but there is no word on them in the discussion. More concordance is expected.

Response: Thank you. We included new points in all manuscript, specially in Discussion and we hope that that clarify all doubts.

Round 2

Reviewer 1 Report

Please see file attached

Author Response

COVER LETTER

We would like to thank you for the opportunity to resubmit the revised version of our manuscript entitled “PREVALENCE OF CONGENITAL INFECTIONS IN NEWBORNS AND UNIVERSAL NEONATAL HEARING SCREENING IN SANTA CATARINA, BRAZIL” for consideration for publication in the Audiology Research. We are thankful to the referees and the Editor for their thoughtful suggestions that helped to strengthen our manuscript. This new version of the manuscript included changes in the manuscript following reviewers’ recommendations. We have addressed all the reviewers’ concerns and provided a detailed response below. To facilitate the process, we have transcribed all questions raised and answered them point by point. In addition, we are attaching a marked version of the manuscript.

Please do not hesitate to contact us if you have any further questions. We look forward to hearing back from you soon.

PREVALENCE OF CONGENITAL INFECTIONS IN NEWBORNS AND UNIVERSAL NEONATAL HEARING SCREENING IN SANTA CATARINA, BRAZIL

The article has really  improved but, I am afraid, I have a few  remarks

Response: Thank you. We appreciate your positive feedback. Thank you for the opportunity to submit the revised version of our manuscript. We have addressed all the comments bellow.

Material and methods

Lines 91-93“In presence of RIHL, the newborns who “passed” in the AABR were referred for  hearing monitoring in primary care, and those who “failed”, were referred for retesting  in the state Outpatient Hearing Health Service “

What do  you mean for retesting? Just  repeat the test or a complete audiological evaluation?

It would also useful to explain which professional make the screening test:  a nurse? An obstetrician? 

AS this article is concerning newborn hearing screening, do  you  think  it could be useful  to know the protocol used: a two  stage protocol ?

Response: Thank you. According to the protocol followed in Brazil (DATAN), newborns who have RIHL and fail the screening are retested once more outside the hospital with the AABR and, if the failure persists, they undergo a complete audiological evaluation for audiological diagnosis. All stages of screening and audiological evaluation are carried out by audiologists and, if it is necessary to confirm the diagnosis of hearing loss, a multidisciplinary team is involved, including the otorhinolaryngologist. This information was included in manuscript.

Discussion

“Despite the prevalence of risk indicators, observed in the present study, only 1.1% of the  newborns failed the hearing screening. However, since collected data refer to a single  moment in the hearing health program, that is, the UNHS, it is not possible to know  whether the diagnosis of hearing loss was confirmed. Due to the presence of RIHL, hearing loss can develop later in many cases and, therefore, the longitudinal follow-up of  these children is fundamental. In addition, those newborns who have failed the UNHS  should be retested as recommended [19-22]. Even those who have passed the UNHS but have RIHL must be monitored in terms of hearing and language development [10,21] “

This paragraph is not  clear. The prevalence of hearing loss has been reported in many articles so it is not unespected that all children with RIHL FAIL THE SCREENING

I suggest to  discuss your results compared to other experiences in which prevalence of congenital infections is similar .

What  do  you think?

Response: Thank you. We agree with you and we try to make the sentence clearer.

Reviewer 2 Report

Thank you for carefully adressing the comments. The revised version is perceived more consistently.

Author Response

COVER LETTER

We would like to thank you for the opportunity to resubmit the revised version of our manuscript entitled “PREVALENCE OF CONGENITAL INFECTIONS IN NEWBORNS AND UNIVERSAL NEONATAL HEARING SCREENING IN SANTA CATARINA, BRAZIL” for consideration for publication in the Audiology Research. We are thankful to the referees and the Editor for their thoughtful suggestions that helped to strengthen our manuscript. This new version of the manuscript included changes in the manuscript following reviewers’ recommendations. We have addressed all the reviewers’ concerns and provided a detailed response below. To facilitate the process, we have transcribed all questions raised and answered them point by point. In addition, we are attaching a marked version of the manuscript.

Please do not hesitate to contact us if you have any further questions. We look forward to hearing back from you soon.

Response: Thank you. We appreciate your positive feedback. 

Round 3

Reviewer 1 Report

The authors answered my questions